# Neferine, an Alkaloid from Lotus Seed Embryos, Exerts Antiseizure and Neuroprotective Effects in a Kainic Acid-Induced Seizure Model in Rats

**DOI:** 10.3390/ijms23084130

**Published:** 2022-04-08

**Authors:** Tzu-Yu Lin, Chih-Yu Hung, Kuan-Ming Chiu, Ming-Yi Lee, Cheng-Wei Lu, Su-Jane Wang

**Affiliations:** 1Department of Anesthesiology, Far-Eastern Memorial Hospital, New Taipei City 22060, Taiwan; drlin1971@gmail.com; 2Department of Mechanical Engineering, Yuan Ze University, Taoyuan 32003, Taiwan; 3School of Medicine, Fu Jen Catholic University, New Taipei City 24205, Taiwan; h.chihyu@gmail.com; 4Cardiovascular Center, Division of Cardiovascular Surgery, Far-Eastern Memorial Hospital, New Taipei 22060, Taiwan; chiu9101018@gmail.com (K.-M.C.); mingyi.lee@gmail.com (M.-Y.L.); 5Department of Electrical Engineering, Yuan Ze University, Taoyuan 32003, Taiwan; 6Research Center for Chinese Herbal Medicine, College of Human Ecology, Chang Gung University of Science and Technology, Taoyuan 33303, Taiwan

**Keywords:** neferine, anti-seizure, neuroprotection, antiinflammation, NLRP3 inflammasome, kainic acid, hippocampus

## Abstract

Current anti-seizure drugs fail to control approximately 30% of epilepsies. Therefore, there is a need to develop more effective anti-seizure drugs, and medicinal plants provide an attractive source for new compounds. This study aimed to evaluate the possible anti-seizure and neuroprotective effects of neferine, an alkaloid from the lotus seed embryos of *Nelumbo nucifera*, in a kainic acid (KA)-induced seizure rat model and its underlying mechanisms. Rats were intraperitoneally (i.p.) administrated neferine (10 and 50 mg/kg) 30 min before KA injection (15 mg/kg, i.p.). Neferine pretreatment increased seizure latency and reduced seizure scores, prevented glutamate elevation and neuronal loss, and increased presynaptic protein synaptophysin and postsynaptic density protein 95 expression in the hippocampi of rats with KA. Neferine pretreatment also decreased glial cell activation and proinflammatory cytokine (interleukin-1β, interleukin-6, tumor necrosis factor-α) expression in the hippocampi of rats with KA. In addition, NOD-like receptor 3 (NLRP3) inflammasome, caspase-1, and interleukin-18 expression levels were decreased in the hippocampi of seizure rats pretreated with neferine. These results indicated that neferine reduced seizure severity, exerted neuroprotective effects, and ameliorated neuroinflammation in the hippocampi of KA-treated rats, possibly by inhibiting NLRP3 inflammasome activation and decreasing inflammatory cytokine secretion. Our findings highlight the potential of neferine as a therapeutic option in the treatment of epilepsy.

## 1. Introduction

Epilepsy is a chronic neurological disorder characterized by recurrent, spontaneous, and unpredictable seizures and affects up to 70 million people worldwide [1]. Currently available anti-seizure drugs (ASDs), which are the main treatment for epilepsy, mainly act by blocking Na^+^ channels, inhibiting glutamatergic transmission, or enhancing GABAergic transmission [2]. However, long-term treatment with these ASDs is often accompanied by many side effects, and approximately 30% of patients with epilepsy do not respond to these drugs [3,4]. Therefore, there is still a need to search for new, more effective and safer anti-seizure medications. In this context, medicinal plants can potentially play an important role in the development of new ASDs since a diverse group of plant-derived compounds have shown promising anticonvulsant effects in different seizure models [5,6,7,8,9].

Neferine is an alkaloid extracted from the seeds of lotus, *Nelumbo nucifera* Gaertn [10,11], and has been reported to exhibit antioxidant, anti-inflammatory, antithrombotic, antidiabetic, cardioprotective, and antitumor properties [12,13,14,15,16,17]. In addition, its neuroprotective activity in animal models was also noted. It attenuates brain damage, improves memory and learning abilities, and has antidepressant action [18,19,20,21,22,23]. However, there is no scientific evidence regarding the anti-seizure effect of neferine. The general objective was therefore to evaluate the action of neferine on kainic acid (KA)-induced seizures in rats and its underlying mechanisms. KA is an analog of glutamate, and the systemic administration of kA to animals causes behavioral seizures and neuropathological lesions which are similar to human epilepsy [24,25]. Therefore, the KA-induced seizure model has been widely recognised as an important experimental model in the research of epilepsy and drug discovery. Our results indicate that pretreatment with neferine significantly attenuated seizure activity, neurotoxicity, and neuroinflammation induced by KA. These beneficial effects are likely mediated by downregulating glutamatergic hyperactivity and the NOD-like receptor 3 (NLRP3)-mediated inflammatory signaling pathway. In fact, experimental models have demonstrated the implication of increased glutamatergic activity and NLRP3 inflammasome activation in the mechanism of epileptogenesis [26,27,28]. Our findings reveal for the first time that neferine possesses potential as an antiepileptogenic agent as well as efficacy in the management of epilepsy.

## 2. Results

### 2.1. Pretreatment with Neferine Attenuates Seizure Activity in KA-Treated Rats

The experimental process is shown in Figure 1A. We first evaluated the anti-seizure effect of neferine. For this, the effect of neferine on seizure activity induced by KA (15 mg/kg, intraperitoneally (i.p.)) was investigated by administering neferine (10 or 50 mg/kg, i.p.) 30 min before KA injection. The dose and schedule of administration were chosen based on our pilot study and others [29,30]. Data analysis by Kruskal–Wallis test yielded statistically significant differences between the tested groups of animals (seizure latency, statistic = 21.9, eta squared = 0.42, *p* < 0.001, *n* = 9–13/group, Figure 1B; seizure score, statistic = 15.6, eta squared = 0.46, *p* < 0.001, *n* = 11–13/group, Figure 1C). Dunn’s post-hoc test showed that pre-exposure to neferine in KA-treated rats significantly delayed seizure onset and decreased seizure severity compared with the KA alone group (*p* < 0.01; Figure 1B,C). In addition, KA caused 32% mortality in injected rats, and this phenomenon was decreased by 15–18% by neferine pretreatment.

### 2.2. Pretreatment with Neferine Alleviates Neuronal Damage in the Hippocampi of KA-Treated Rats

Morphological changes in the hippocampus, particularly the CA1 and CA3 regions, have been shown in KA-induced seizure rats [8,31]. To evaluate whether neferine has a neuroprotective effect in KA-treated rats, we performed Nissl staining at 72 h after KA injection to observe the neuronal morphology in the hippocampus. As shown in Figure 2A, neurons in the hippocampi of rats in the control group (rats that received dimethylsulfoxide (DMSO)) were regularly arranged. However, rats in the KA group exhibited disorganized neurons and a considerable loss of neurons in the CA1 and CA3 regions. Neferine pretreatment alleviated these phenomena. One-way ANOVA yielded statistically significant differences between the tested groups of animals (CA1, F(3, 19) = 48.4, eta squared = 0.88, *p* < 0.001; CA3, F(3, 19) = 70.6, eta squared = 0.91, *p* < 0.001; *n* = 5–6/group). Tukey post hoc test showed that the number of live neurons in the CA1 and CA3 regions decreased significantly in the KA-treated group compared with the control group (*p* < 0.001). Pre-exposure to neferine in KA-treated rats significantly prevented this reduction compared to the KA alone group (*p* < 0.001) (Figure 2B). In addition, the number of neuronal nuclei (NeuN)-positive cells, which was measured by using NeuN immunohistochemistry in the CA1 and CA3 regions, was counted to investigate neuronal death in the hippocampus (Figure 2A,C). One-way ANOVA yielded statistically significant differences between the tested groups of animals (CA1, F(3, 21) = 12.9, eta squared = 0.65, *p* < 0.01; CA3, F(3, 21) = 25.9, eta squared = 0.79, *p* < 0.001; *n* = 5–7/group; Figure 2C). Tukey post hoc test showed that the number of NeuN-positive cells in the CA1 and CA3 regions decreased significantly in the KA-treated group compared with the control group (*p* < 0.01). Pre-exposure to neferine in KA-treated rats increased the number of NeuN-positive cells compared to the KA alone group (*p* < 0.001) (Figure 2C). The expression levels of NeuN in the hippocampus were also examined using Western blotting (Figure 2D). One-way ANOVA yielded statistically significant differences between the tested groups of animals (F(3, 16) = 58.8, eta squared = 0.92, *p* < 0.001; *n* = 5/group). Tukey post hoc test indicated that the expression of NeuN in the hippocampi of KA-treated rats was significantly lower than that in the control group (*p* < 0.001), whereas neferine pretreatment significantly increased the expression of NeuN compared with the KA group (*p* < 0.001) (Figure 2D).

### 2.3. Pretreatment with Neferine Decreases Glutamate Elevation and Increases the Expression of Synaptic Proteins in the Hippocampi of KA-Treated Rats

Excess glutamate and synaptic dysfunction are observed in the hippocampi of KA-treated rats and are associated with neuronal death [32,33]. Accordingly, we performed high-performance liquid chromatography (HPLC) and Western blot analysis at 72 h after KA injection to evaluate the effect of neferine pretreatment on the concentration of glutamate and the expression of presynaptic protein synaptophysin and postsynaptic density protein 95 (PSD95) in the hippocampus (Figure 3). One-way ANOVA revealed statistically significant differences between the tested groups of animals (glutamate level, F(3, 16) = 33.5, eta squared = 0.68, *p* < 0.001, *n* = 5/group, Figure 3A; synaptophysin, F(3, 16) = 32.8, eta squared = 0.86, *p* < 0.001; *n* = 5/group; PSD-95, F(3, 16) = 24.4, eta squared = 0.82, *p* < 0.001; *n* = 5/group; Figure 3B). Tukey post hoc test showed that the concentration of glutamate was increased (*p* < 0.001), whereas the expression levels of synaptophysin (*p* < 0.001) and PSD95 (*p* < 0.001) were decreased in the hippocampi of rats in the KA-treated group compared with the control group. Pre-exposure to neferine in KA-treated rats decreased glutamate levels (*p* < 0.001) but increased synaptophysin and PSD95 expression in comparison with the KA alone group (*p* < 0.001) (Figure 3A,B).

### 2.4. Pretreatment with Neferine Suppresses the Activation of Glial Cells in the Hippocampi of KA-Treated Rats

To gain further insight into the anticonvulsant activity of neferine, we analyzed the activation of glial cells, including astrocytes and microglia, which are known to be involved in the mechanism of epileptogenesis [34]. We assessed the effects 72 h after KA injection by measuring the protein expression level of glial fibrillary acidic protein (GFAP) for astrogliosis and CD11b for microgliosis within the hippocampus (Figure 4). One-way ANOVA yielded statistically significant differences between the tested groups of animals (GFAP, F(3, 16) = 121.1, eta squared = 0.96, *p* < 0.001; *n* = 5/group; CD11b, F(3, 16) = 17.8, eta squared = 0.77, *p* < 0.001; *n* = 5/group; Figure 4). Tukey post hoc test indicated that KA induced an increase in the expression levels of the GFAP and CD11b proteins in the hippocampus compared with the control group (*p* < 0.001). Pre-exposure to neferine in KA-treated rats significantly decreased the expression of these two reactive gliosis biomarkers in the hippocampus compared with the KA-treated group (*p* < 0.001; Figure 4). Immunohistochemistry results further revealed that KA-induced gliosis occurred in the hippocampus, as illustrated by the higher numbers of GFAP- and CD11b-positive cells in the CA1 and CA3 regions than in the control group (GFAP: CA1, F(3, 20) = 27.1, eta squared = 0.8, *p* < 0.001; CA3, F(3, 20) = 16.9, eta squared = 0.72, *p* < 0.001; *n* = 4–8/group, Figure 5A,B; CD11b: CA1, F(3, 16) = 123.1, eta squared = 0.96, *p* < 0.001; CA3, F(3, 16) = 130.7, eta squared = 0.96, *p* < 0.001; *n* = 5/group; Figure 5C,D). Compared with the KA group, KA-treated rats pre-exposed to neferine had reduced numbers of GFAP- and CD11b-positive cells in the CA1 and CA3 regions (*p* < 0.001; Figure 5B,D).

### 2.5. Pretreatment with Neferine Decreases the Expression of Proinflammatory Cytokines in the Hippocampi of KA-Treated Rats

As KA-induced gliosis is closely associated with the production of proinflammatory cytokines [35,36], we analyzed the protein expression levels of the proinflammatory cytokines interleukin-1β (IL-1β), interleukin-6 (IL-6), and tumor necrosis factor-α (TNF-α) at 72 h after KA injection in the rat hippocampus using Western blot analysis. One-way ANOVA revealed statistically significant differences between the tested groups of animals (IL-1β, F(3, 16) = 187.1, eta squared = 0.97, *p* < 0.001; *n* = 5/group; IL-6, F(3, 16) = 1651.8, eta squared = 0.99, *p* < 0.001; *n* = 5/group; TNF-α, F(3, 16) = 634.1, eta squared = 0.99, *p* < 0.001; *n* = 5/group; Figure 6). Tukey’s post hoc test indicated that KA-induced an increase in the expression levels of IL-1β, IL-6, and TNF-α proteins in the hippocampus compared with the control group (*p* < 0.001). Pre-exposure to neferine in KA-treated rats significantly decreased the protein expression levels of these three proinflammatory cytokines in the hippocampus compared with the KA-treated group (*p* < 0.001).

### 2.6. Pretreatment with Neferine Decreases the Activation of the NLRP3 Inflammasome Pathway in the Hippocampi of KA-Treated Rats

To further explore whether the NLRP3 inflammasome pathway is involved in the anti-inflammatory action of neferine, we assessed the expression levels of the NLRP3 inflammasome signaling-related proteins—NLRP3, caspase-1, and interleukin-18 (IL-18) in the hippocampus at 72 h after KA injection. Western blotting analysis with one-way ANOVA revealed statistically significant differences in the expression levels of NLRP3 (F(3, 16) = 463.9, eta squared = 0.99, *p* < 0.001; *n* = 5/group), caspase-1 (F(3, 16) = 1651.8, eta squared = 0.99, *p* < 0.01; *n* = 5/group), and IL-18 (F(3, 16) = 634.1, eta squared = 0.93, *p* < 0.001; *n* = 5/group; Figure 7). Tukey post hoc test indicated that KA induced substantial increase in the protein expression levels of NLRP3, caspase-1, and IL-18 in the hippocampus compared with the control group (*p* < 0.001). Pre-exposure to neferine in KA-treated rats significantly decreased NLRP3, caspase-1, and IL-18 expression in the hippocampus compared with only KA-treated group (*p* < 0.001).

## 3. Discussion

Epilepsy is a chronic neurological disease that affects millions of people. Current therapy suffers from various limitations, including low efficacy and serious side effects [1,3]. A solution to this problem is to seek out a novel anti-seizure drug from medicinal plants. In fact, plant-derived compounds are known to exhibit anti-seizure activity in animal models with fewer side effects [37]. Therefore, this study aimed to evaluate the effect of neferine pretreatment on seizures induced by the glutamate analog KA in rats. In this model, which mimics the main features of human epilepsy [24], the systemic administration of KA to animals induces tonic-clonic seizures and causes severe loss of neurons and synaptic dysfunction in the central nervous system, particularly in the hippocampus [28,31,32,38]. In the present study, epileptic animals were pretreated with neferine. The results showed that neferine significantly increased seizure latency and decreased seizure severity, demonstrating its anti-seizure activity. Neferine pretreatment also decreased neuronal loss in the CA1 and CA3 regions of the hippocampus, which is prone to damage by KA [39,40]. Furthermore, the decreases in the expression levels of the synaptic proteins synaptophysin and PSD95 in the hippocampi of KA-treated rats were reversed by neferine pretreatment. In fact, downregulation of synaptophysin and PSD95 in the hippocampi of rats treated with KA was associated with the observed hippocampal neuronal loss [31,32]. Therefore, we suggest that neferine can prevent KA-induced seizures and have preventive effects on nerve cell damage and synaptic dysfunction in the hippocampus. Although the ability of neferine to pass the blood-brain barrier remains to be elucidated, its neuroprotective effect on the central nervous system (CNS) has been reported in several animal models of neurological disorders, including cerebral ischemia, Alzheimer’s disease, and depression [18,19,21,22,23].

Various mechanisms have been investigated to understand the etiopathology of KA-seizures; excessive glutamate—a key neurotransmitter in the CNS—has been proposed as one of the main underlying mechanisms [33,41]. Accordingly, the concentration of glutamate in the hippocampus was determined to elucidate the possible mechanism of the anti-seizure and neuroprotective actions of neferine in our study. Consistent with previous studies, KA-treated rats showed a significant increase in hippocampal glutamate concentrations. Interestingly, in KA-treated rats, neferine pretreatment significantly reduced glutamate levels in the hippocampus, suggesting that its anti-seizure and neuroprotective effects might result from the attenuation of glutamatergic hyperactivity. This suggestion was supported by our previous study showing the ability of neferine to inhibit glutamate release from rat cortical nerve terminals [42]. In addition to glutamate, however, further evaluation of the influence of neferine on other neurotransmitter systems, such as γ-aminobutyric acid (GABA) and serotonin, which have been suggested to be involved in epilepsy disorders [43,44], is still warranted.

Suppression of neuroinflammation in the hippocampus of KA-treated rats was another beneficial effect exerted by neferine. Numerous clinical studies and animal experiments have confirmed that the inflammatory response is linked to neuronal death and epileptogenesis [30,42,43]. Glial cells, including astrocytes and microglia, in the CNS, play a key role in the inflammatory reaction. Activation of these cells has been observed in the hippocampi of rats with epilepsy [8,45]. In vivo studies also proved that activated glial cells produce large amounts of proinflammatory cytokines such as TNF-α, IL-1β, and IL-6, which, in turn, cause neuronal damage in the hippocampus of rats with epilepsy [46]. In line with previous studies [8,27,32], we found that treatment of the rats with KA caused significant brain neuroinflammation, as evidenced by increased expression levels of GFAP (an astrocytic marker) and CD11b (a microglial marker), glial cell activation, and proinflammatory cytokine overproduction in the hippocampi of KA-treated rats. These findings suggest the important role of neuroinflammation in the pathophysiology of KA-induced seizures. Additionally, in line with previous studies that have demonstrated that neferine has anti-inflammatory activity [15,16], we found that neferine pretreatment caused significant suppression of neuroinflammation in KA-induced seizure rats, as evidenced by reduced KA-induced gliosis and proinflammatory cytokine production in the hippocampus. Therefore, the ameliorative effect of neferine on inflammation in the hippocampus might participate in its anti-seizure and neuroprotective effects in KA-treated rats.

Previous studies have highlighted the importance of the NLRP3 inflammasome in neuroinflammatory processes during epileptogenesis [27,47,48,49]. The NLRP3 inflammasome is a multiprotein complex that results from the assembly of NLRP3, apoptosis-associated speck-like protein (ASC), and pro-caspase-1. Activation of the NLRP3 inflammasome causes the activation of caspase-1, which cleaves the inactive proinflammatory cytokines pro-IL-1β and pro-IL-18 into mature IL-1β and IL-18 [50,51]. In animal and clinical epilepsy-related research, increased levels of the NLRP3 inflammasome and neuroinflammatory cytokines have been detected in hippocampal tissues [24,27,46]. Moreover, numerous studies have reported that inhibiting NLRP3 activation and inflammatory cytokine secretion decreases the loss of neurons and the severity of seizures [52,53]. In the present study, the results revealed increases in NLRP3, caspase-1, IL-1β, and IL-18 expression levels in the hippocampi of rats with KA, phenomena that were significantly reversed by neferine pretreatment. Therefore, the suppression of NLRP3 inflammasome activation and downstream inflammatory cytokine secretion by neferine may play a role, at least in part, in the amelioration of seizures and neuronal death in KA-treated rats. Our findings are similar to previous results in which neferine was found to exert neuroprotective effects and effectively suppress NLRP3 inflammatory activation [23,48]. However, how neferine affects NLRP3 inflammasome signaling is still unclear. Further research is necessary to clarify the precise mechanisms.

In the present study, neferine, at 10 and 50 mg/kg, exerts similar effects in all analyzed parameters. We infer that the preventive effects of neferine in KA-induced seizure rats might also occur in a lower dose range. Although we did not examine the effect of neferine at the relatively low dose in this study, our finding is consistent with previous studies, which have reported that neferine at a dose of 12–50 mg/kg attenuates ischemia-induced brain damage in rats [18,19,23].

## 4. Materials and Methods

### 4.1. Chemicals and Their Sources

Neferine (purity > 98%, CAS No. 2292-16-2) was purchased from ChemFaces (Wuhan, Hubei, China). KA, DMSO, and other general reagents were purchased from Sigma-Aldrich (St. Louis, MO, USA).

### 4.2. Animals and Ethics

Male Sprague-Dawley rats weighing 150–200 g were purchased from BioLASCO (Taipei, Taiwan) and were housed under controlled conditions (18–25 °C; 50–60% humidity; 12 h light/dark cycle) with food pellets and water available ad libitum. The animals were used after a one-week period of quarantine and acclimatization. All experimental protocols in this study were approved by the Institutional Animal Care and Use Committee of Fu Jen Catholic University (A10911, 26 August 2020), and the animals were cared for in accordance with the National Institutes of Health Guide for the Care and Use of Laboratory Animals.

### 4.3. Seizure Induction and Animal Grouping

Seizures were induced by i.p. injection of rats with KA (15 mg/kg), which was dissolved in distilled water. The dose of KA was based on previous studies in KA-treated seizure rats [9,54]. After KA injection, the rats were placed individually in cages and observed for 3 h for the development of seizures. The latency to and intensity of seizures were recorded. Seizure stages were scored according to the scale devised by Racine et al., (1972) [55]: stage 1, facial clonus; stage 2, head nodding; stage 3, forelimb clonus; stage 4, forelimb clonus with rearing; and stage 5, rearing, jumping, and falling. The rats were divided into four groups, including the control (rats-received DMSO), KA, neferine 10 mg/kg + KA, and neferine 50 mg/kg + KA. Neferine was dissolved in a saline solution containing DMSO 1% and administrated (30 μL, i.p.) 30 min before KA injection. 72 h after KA injection, the rats were euthanized using CO_2_ or cervical dislocation to obtain the brain tissue samples. Totally 52 rats were used in this study. The seizure behavior was evaluated in all rats. Immunohistochemistry was performed on the fixed brain tissue of 32 rats in four groups (*n* = 5–8 rats/group). Fresh brain tissue of 20 rats was used for HPLC and Western blotting in four groups (*n* = 5 rats/group).

### 4.4. Immunohistochemistry

Seventy-two hours after KA injection, rats were euthanized with CO_2_ then perfused intracardially with ice-cold phosphate-buffered saline (PBS) followed by 4% paraformaldehyde. Brains were removed, fixed in 4% paraformaldehyde for 24 h, then underwent dehydration using 30% sucrose, then finally were cut into 25 μm coronal sections (frozen cryosections). For Nissl staining, the sections were mounted on gelatin-coated slides and air-dried overnight. The slides were rehydrated in distilled water and stained in a 0.1% cresyl violet solution (Sigma-Aldrich, St. Louis, MO, USA) for 20 min. After rinsing with distilled water, the slides were gradually dehydrated with a series of alcohols, cleared in xylene, and cover-slipped. Each stained section was observed with a light microscope BX-50 (Olympus, Tokyo, Japan) to assess the degree of nerve cell loss within the hippocampus.

NeuN immunofluorescence staining was performed as described previously [8]. The sections were incubated in phosphate-buffered saline containing 10% normal goat serum for 1 h to block nonspecific antibody binding. The sections were then incubated with NeuN antibody (1:500, Merck Millipore, Billerica, MA, USA) at 4 °C overnight. The next day, the IgG-DyLight 594 (1:1000, Vector Laboratories, Burlingame, CA, USA) was applied for 1 h at room temperature. Finally, 4,6-diamino-2-phenylindole (DAPI) was used for nucleus staining, and sections were mounted onto gelatin-coated glass slides and coverslipped with a mounting medium.

For GFAP and CD11b staining, the sections were treated with primary antibodies (GFAP, 1:1000; Cell Signaling, Beverly, MA, USA; CD11b, 1:500; Abcam, Cambridge, UK) at 4 °C overnight. The sections were then incubated with goat biotinylated anti-mouse secondary antibodies (1:200; Vector Laboratories, Burlingame, CA, USA) for 2 h and subsequently incubated with ExtrAvidin peroxidase (1:1000) for 1h at room temperature. After rinsing in 0.1 M phosphate-buffered saline for 20 min, the sections were reacted with 0.025% 3,3-diaminobenzidine tetrahydrochloride (DAB) solution in phosphate-buffered saline containing 0.0025% hydrogen peroxide for 6 min. The sections were then mounted onto gelatin-coated glass slides and coverslipped with a mounting medium.

Images were captured at 4×, 10×, and 20× magnification using a fluorescence microscope (Zeiss Axioskop 40, Göttingen, Lower Saxony, Germany) under identical settings for each channel. The number of surviving neurons and NeuN-, GFAP-, or CD11b-positive cells were counted in an area of 255 µm × 255 µm of the hippocampal CA1/CA3 in 6 to 8 randomly chosen sections from each animal and averaged for each animal using ImageJ software (Synoptics, Cambridge, Cambridgeshire, UK).

### 4.5. Quantification of Brain Glutamate Content

The measurement of glutamate levels in the hippocampus is based on a previously described method [54]. In brief, rats were euthanized by cervical dislocation 72 h after KA injection. The hippocampi were rapidly dissected, blotted, weighed, put into 1 mL ice-cold perchloric acid, homogenized, and centrifuged at 10,000× *g* for 10 min. Subsequently, the supernatant (10 μL) was filtered and injected directly into an HPLC system with electrochemical detection (HTEC500, Eicom, Kyoto, Japan). The relative free glutamate concentration was determined based on peak areas by an external standard method. Glutamate content was expressed as μM/mg of brain tissue.

### 4.6. Western Blotting

At 72 h after KA injection, the rats were euthanized by cervical dislocation, and the hippocampus was rapidly dissected. Dissected hippocampi were immediately immersed individually in a lysis buffer and sonicated for 10 s. Immunoblotting was performed as described previously [8,54]. Briefly, equal amounts of protein were separated with sodium dodecyl sulfate (SDS)-polyacrylamide gel electrophoresis (Bio-Rad, Hercules, CA, USA) and transferred onto polyvinylidene fluoride membrane by using Trans-Blot Turbo Transfer System (Bio-Rad). The membranes were then incubated overnight at 4 °C with anti-β-actin (1:5000; Cell Signaling, Beverly, MA, USA), anti-NeuN (1:3000; Merck Millipore, Billerica, MA, USA), anti-GFAP (1:5000; Cell Signaling, Beverly, MA, USA), anti-synaptophysin (1:5000; Cell Signaling, Beverly, MA, USA), anti-PSD95 (1:1000; Abcam, Cambridge, Cambridgeshire, UK), anti-CD11b (1:8000; Abcam, Cambridge, Cambridgeshire, UK), anti-IL-1β (1:500; Cell Signaling, Beverly, MA, USA), anti-IL-6 (1:100; Abcam, Cambridge, Cambridgeshire, UK), anti-TNF-α (1:200; Abcam, Cambridge, Cambridgeshire, UK), anti-IL-18 (1:2000; Abcam, Cambridge, Cambridgeshire, UK), anti-NLRP3 (1:3000; Abcam, Cambridge, Cambridgeshire, UK), or anti-cleaved caspase-1 (1:2000; Cell Signaling, Beverly, MA, USA) antibodies. The membrane was then incubated with a horseradish peroxidase-conjugated secondary antibody (1:5000; Genetex, Zeeland, MI, USA) for 2 h at room temperature. After extensive washing, the bands were developed using enhanced chemiluminescence (Amersham ECL Detection Reagents; Cytiva, Little Chalfont, Buckinghamshire, UK). The densities of the bands were measured and analyzed using ImageJ software (Synoptics, Cambridge, Cambridgeshire, UK). To determine the relative band density ratio, all values were normalized against β-actin.

### 4.7. Data Analysis and Statistics

The data were presented as means ± standard error of the mean (SEM) per group. Data were checked for normal distribution using the Kolmogorov-Smirnov test. Data for Figure 1 were analyzed using Kruskal-Wallis with Dunn post hoc test. Data for Figure 2, Figure 3, Figure 4, Figure 5, Figure 6 and Figure 7 were analyzed using a one-way analysis of variance (ANOVA) with Tukey’s post hoc test. Eta squared was used to assess the effect size [56]. All analyses were conducted using GraphPad Prism 8 (version 8.4.3, GraphPad Software, San Diego, CA, USA). *p* < 0.05 was considered significant.

## 5. Conclusions

This study introduces new information on the ameliorative effect of neferine on KA-induced seizures in rats. Neferine reduced seizure severity, attenuated neuronal loss, inhibited excess glutamate, and suppressed inflammation in the hippocampi of seizure rats. Suppression of NLRP3 inflammasome activation and inflammatory cytokine secretion is the main mechanism underlying the protective effect of neferine. Therefore, our results demonstrated that neferine might be a potentially valuable resource for the prevention and therapy of epilepsy.

## Figures and Tables

**Figure 1 ijms-23-04130-f001:**
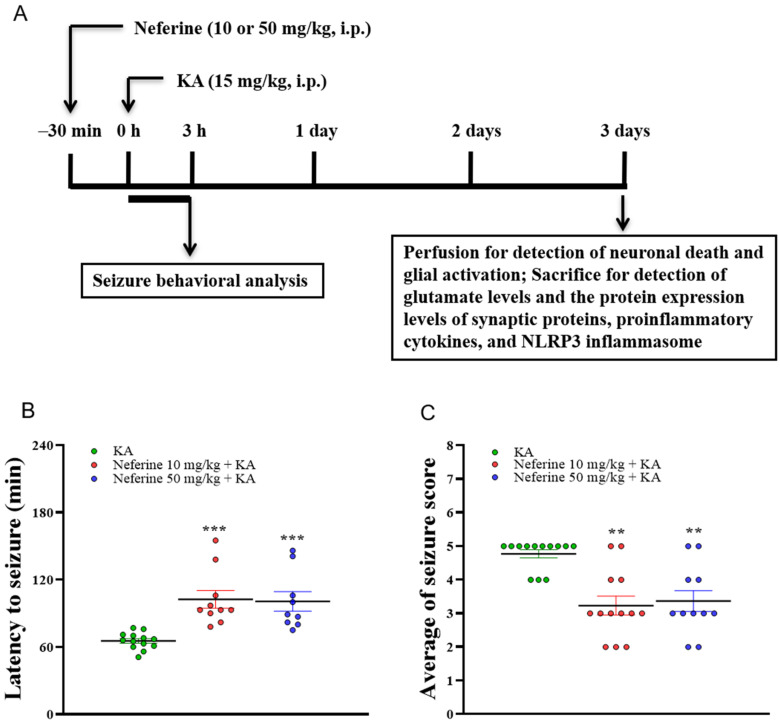
Neferine reduces KA-induced seizure activity. (**A**) Experimental design. (**B**,**C**) Seizure behavior analysis in the different groups (*n* = 9–13 rats/group). Statistical results showed that neferine increased the latency of seizures (**B**) and decreased seizure score (**C**) (one-way ANOVA). ** *p* < 0.01, *** *p* < 0.001 vs. KA-treated group.

**Figure 2 ijms-23-04130-f002:**
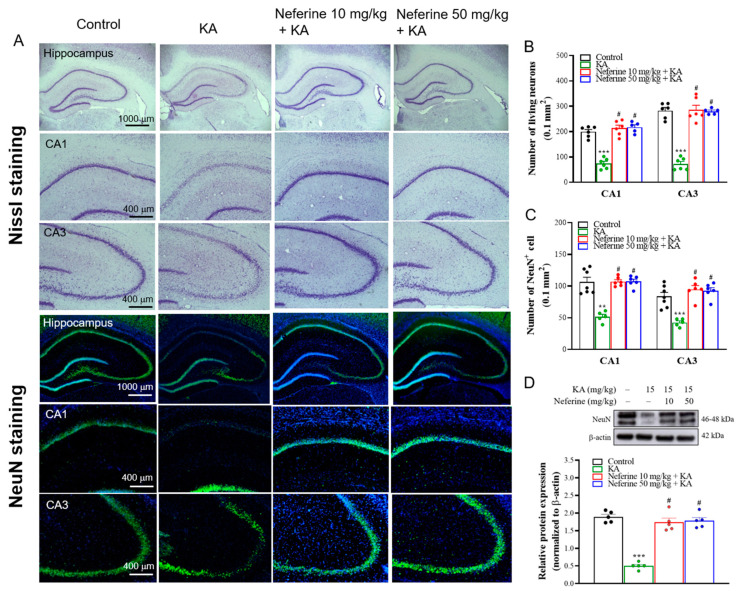
Results of Nissl and NeuN staining in the rat hippocampus. (**A**) Representative images and (**B**,**C**) quantitative data for the number of Nissl- or NeuN-positive hippocampal neurons (*n* = 5–7 rats/group). Neferine increased the numbers of Nissl- or NeuN-positive hippocampal neurons in the CA1 and CA3 areas (one-way ANOVA). (**D**) Representative Western blot images in the different groups and densitometric values for NeuN were normalized to β-actin levels (*n* = 5 rats/group). Statistical results of immunoblot analysis show that neferine increased the band intensities of NeuN (one-way ANOVA). ** *p* < 0.01. *** *p* < 0.05 vs. control group. # *p* < 0.05 vs. KA-treated group.

**Figure 3 ijms-23-04130-f003:**
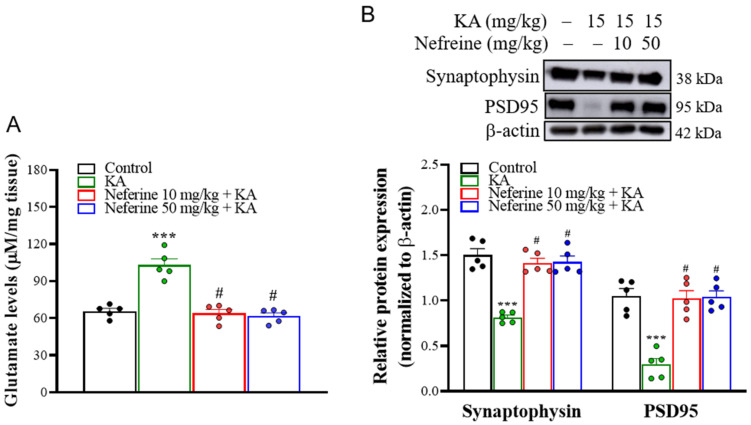
(**A**) HPLC analysis of hippocampal glutamate level in the different groups (*n* = 5 rats/group). Results show that neferine reduced the levels of glutamate (one-way ANOVA). (**B**) Western blot analyses of synaptophysin and PSD95 proteins in rat hippocampal tissue. Representative Western blot images in the different groups and densitometric values for synaptophysin and PSD95 were normalized to β-actin levels (*n* = 5 rats/group). Statistical results of the immunoblot analysis showed that neferine increased the band intensity of synaptophysin and PSD95 (one-way ANOVA). *** *p* < 0.05 vs. control group. # *p* < 0.05 vs. KA-treated group.

**Figure 4 ijms-23-04130-f004:**
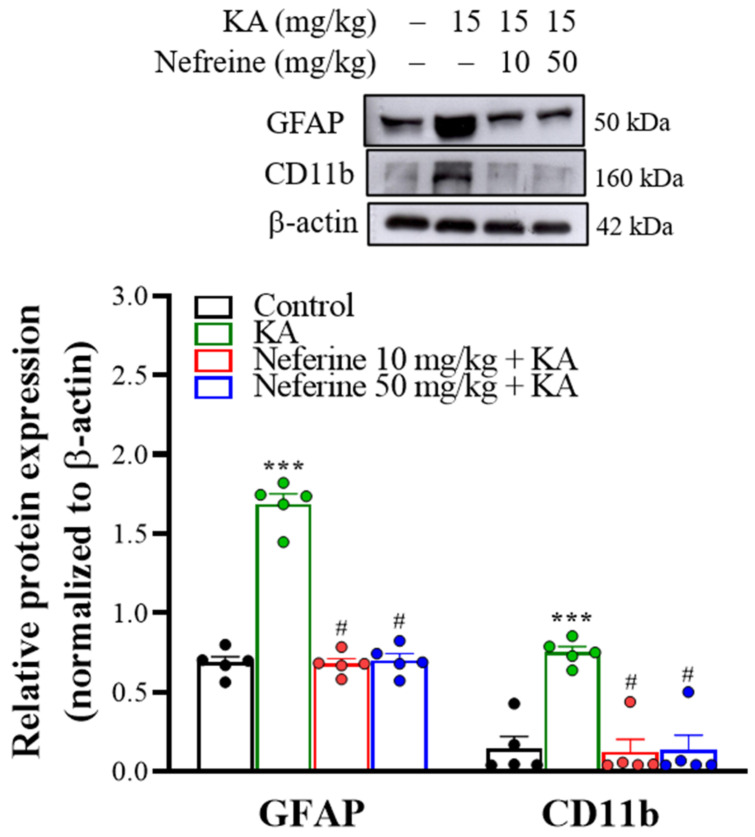
Western blot analyses of GFAP and CD11b proteins in rat hippocampal tissue. Representative images in the different groups and densitometric values for GFAP and CD11b were normalized to β-actin levels (*n* = 5 rats/group). Statistical results of the immunoblot analysis show that neferine reduced the band intensity of GFAP and CD11b (one-way ANOVA). *** *p* < 0.05 vs. control group. # *p* < 0.05 vs. KA-treated group.

**Figure 5 ijms-23-04130-f005:**
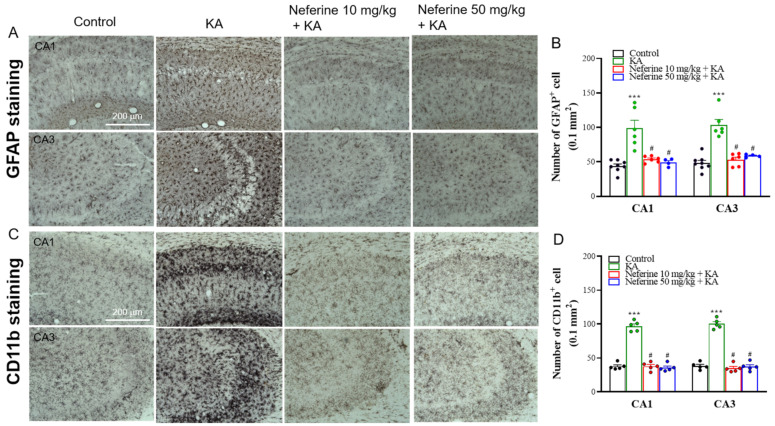
Results of GFAP and CD11b staining in the rat hippocampus. (**A**,**C**) Representative images in the different groups and (**B**,**D**) quantitative data for the number of GFAP- or CD11b-positive hippocampal neurons (*n* = 4–8 rats/group). Neferine decreased the numbers of GFAP- or CD11b-positive hippocampal neurons in the CA1 and CA3 areas (one-way ANOVA). *** *p* < 0.05 vs. control group. # *p* < 0.05 vs. KA-treated group.

**Figure 6 ijms-23-04130-f006:**
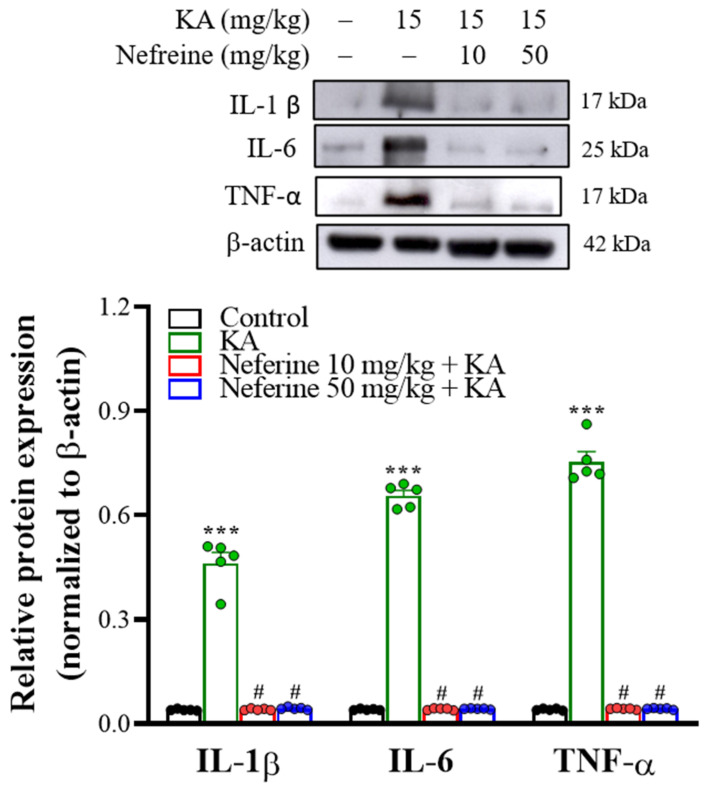
Neferine affects proinflammatory cytokine production in rat hippocampal tissue. Representative Western blot images in the different groups and densitometric values for IL-1β, IL-6, and TNF-α were normalized to β-actin levels (*n* = 5 rats/group). Statistical results of immunoblot analysis showed that neferine reduced the band intensities of IL-1β, IL-6, and TNF-α (one-way ANOVA). *** *p* < 0.05 vs. control group. # *p* < 0.05 vs. KA-treated group.

**Figure 7 ijms-23-04130-f007:**
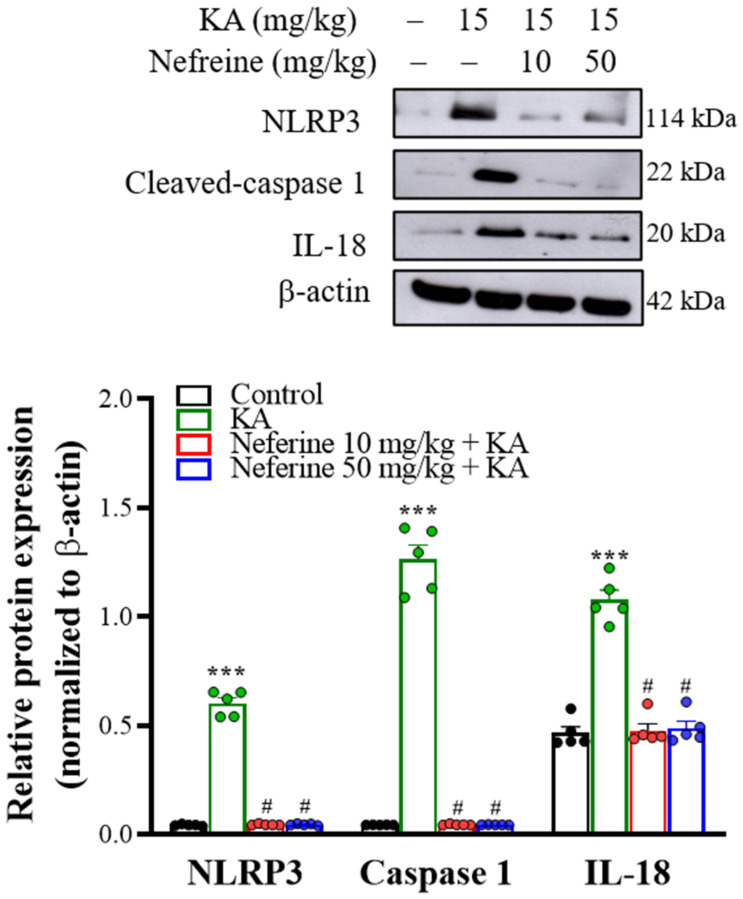
Western blot analyses of NLRP3 inflammasome-related proteins in rat hippocampal tissue. Representative images in the different groups and densitometric values for NLRP3, active caspase-1, and IL-18 were normalized to β-actin levels (*n* = 5 rats/group). Statistical results of the immunoblot analysis show that neferine reduced the band intensity of NLRP3, active caspase-1, and IL-18 (one-way ANOVA). *** *p* < 0.05 vs. control group. # *p* < 0.05 vs. KA-treated group.

## Data Availability

The data presented in this study are available on request from the corresponding author.

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
