# Peer review of "Neferine, an Alkaloid from Lotus Seed Embryos, Exerts Antiseizure and Neuroprotective Effects in a Kainic Acid-Induced Seizure Model in Rats"

_ijms, 2022, doi:10.3390/ijms23084130_

Round 1

Reviewer 1 Report

The authors deal with an up-to-date subject: the benefits of plant extracts in medications. They use a wide variety of techniques; prove their theory in many aspects. They answer in the paper all of my possible questions. The introduction and the discussion are well-written, giving explanation for the main question of the paper. The figures are neat, informative. All the conclusions are supported with well-designed experiments. I accept the paper as it is.

Author Response

Response to reviewer1

ijms-1633899R1

The authors deal with an up-to-date subject: the benefits of plant extracts in medications. They use a wide variety of techniques; prove their theory in many aspects. They answer in the paper all of my possible questions. The introduction and the discussion are well-written, giving explanation for the main question of the paper. The figures are neat, informative. All the conclusions are supported with well-designed experiments. I accept the paper as it is.

 We thank the reviewer for the critical comments.

Reviewer 2 Report

In this manuscript, authors evaluated the molecule neferine as a possible antiseizure medication able to produce anti-neuroinflammatory and neuroprotective effects. My comments are as follows:

  • Authors adopted the kainic acid model of status epilepticus in rats, but no description of this model is found in the introduction (line 59). Authors should refer to the recent work of Costa et al (2020 doi: 10.33594/000000232) to provide some information on the model.
  • Neferine was administered in advance with respect to kainic acid and, as shown by the authors in the figure 1B and 1C, modified the development of status epilepticus. For this reason, the authors’ findings were strongly biased by prevention of the status epilepticus, which is critical to provoke the lesions (see Curia et al 2014, doi: 10.2174/0929867320666131119152201). Since a significant number of animals did not respond to kainic acid (as reported in figure 1C, 4 out of 10 in the 50 mg/kg group, 2 out of 10 in the 10 mg/kg group), these rats have to be discarded from the analysis of damage and neuroinflammation because they did not develop any lesion.
  • Additionally, data presented in figure 1C cannot be analyzed by ANOVA and Tukey, since they were categorical. Authors must use the Kruskal-Wallis test followed by Dunn post hoc test.
  • The lane of beta-actin used in the various western blot is always the same. This should be explained.
  • The manuscript should be re-reviewed after complete correction of the results.

Author Response

Response to reviewer2

ijms-1633899R1

We thank the reviewer for the critical comments and constructive suggestions.

  • Authors adopted the kainic acid model of status epilepticus in rats, but no description of this model is found in the introduction (line 59). Authors should refer to the recent work of Costa et al (2020 doi: 10.33594/000000232) to provide some information on the model.

As suggestion by the reviewer, the sentences KA is an analog of glutamate, and the systemic administration of kA to animals causes behavioral seizures and neuropathological lesions which are similar to human epilepsy [24,25]. Therefore, KA-induced seizure model has been widely recognised as an important experimental model in the research of epilepsy and drug discovery. are added in the introduction (Page 2, Lines 61-65 ).

  • Neferine was administered in advance with respect to kainic acid and, as shown by the authors in the figure 1B and 1C, modified the development of status epilepticus. For this reason, the authors’ findings were strongly biased by prevention of the status epilepticus, which is critical to provoke the lesions (see Curia et al 2014, doi: 10.2174/0929867320666131119152201). Since a significant number of animals did not respond to kainic acid (as reported in figure 1C, 4 out of 10 in the 50 mg/kg group, 2 out of 10 in the 10 mg/kg group), these rats have to be discarded from the analysis of damage and neuroinflammation because they did not develop any lesion.

According this point, Figure 1B and C are modified. In the present study, rats were intraperitoneally (i.p.) administrated neferine (10 and 50 mg/kg) 30 min before KA injection (15 mg/kg, i.p.). In pre-exposure to neferine in KA-treated rats, several rats do not exhibit epileptic behavior showing that neferine inhibits epileptogenesis. Therefore, these rats are also be included in subsequent experiments in the present study. In fact, animal epilepsy behavior and brain injury situations do not necessarily correspond. Hope the reviewer can accept our reply. Your suggestion will help our future experimental design.

Additionally, data presented in figure 1C cannot be analyzed by ANOVA and Tukey, since they were categorical. Authors must use the Kruskal-Wallis test followed by Dunn post hoc test.

As suggestion by the reviewer, figure 1C is analyzed by the Kruskal-Wallis test followed by Dunn post hoc test. The sentence is modified in the result section (Page 2, Lines 78-82). In addition, the sentence or Kruskal-Wallis test with Dunn post hoc testis added in the method section (Page 18, Line 406-407).

  • The lane of beta-actin used in the various western blot is always the same. This should be explained.

According to this point, Figure 3, 4, 6, and 7 are modified.

  • The manuscript should be re-reviewed after complete correction of the results.

The manuscript is reviewed.

Reviewer 3 Report

Comments for: “Neferine, an Alkaloid from Lotus Seed Embryos, Exerts Anticonvulsant and Neuroprotective Effects in a Kainic Acid-induced Seizure Model in Rats”. The aim of this study was to investigate the effects of neferine administration in the kainic acid induced seizure model.

From my point of view, several concerns of this paper are related to methods that do not allow to draw conclusions.

In details:

  • Throughout the text, inadequate terms are used such as for instance "antiepileptic drugs" and many more. The authors should use an up-to-date terminology.
  • Authors should justify the doses of neferine used. Likewise, in agreement with PK of neferine, also the time of administration should be justified.
  • The method used to solubilize neferine should be better described. How much DMSO was administered for rat? DMSO has demonstrated antioxidant, neuroprotective and cryopreservative effects. Furthermore, it has been demonstrated that DMSO, in a dose dependent manner, has different effects on seizures. Finally, I would like to understand whether the vehicle has any effect.
  • Authors have written in methods: “The rats were divided into four groups (5 rats per group)……….”. I have difficult to understand the number of rats used.
  • Authors should report the mortality rate in rats after Kainate injection?
  • The effect size should also be reported. Likewise, the statistics section should be better described. ANOVA was used to analyze what data? Kruskal-Wallis test was used to analyze what? Why these two tests?
  • Authors in the discussion affirmed that: “this study aimed to evaluate the effect of neferine pretreatment on convulsions during epileptogenesis induced by the glutamate analog KA in rats” As previously reported the epileptogenic period in this model last about 30 days. By virtue of this, this study did not cover the entire period of epileptogenesis but only the early phase of the latent period.
  • Similarly authors concluded that:” Therefore, neferine could confer beneficial effects against epileptogenesis…..” From my point of view, the experimental design of this study do not permit to draw a similar conclusion.
  • I would like to understand why similar effects by different doses of neferine. This should be discussed.
  • Authors affirmed that: “Therefore, we suggest that neferine might antagonize KA-induced seizures……. “. From my point of view, this sentence is too speculative.

Author Response

Response to reviewer 3

ijms-1633899R2

We thank the reviewer for the critical comments and constructive suggestions.

  • Throughout the text, inadequate terms are used such as for instance "antiepileptic drugs" and many more. The authors should use an up-to-date terminology.

The word "antiepileptic" is modified to "anti-seizure" (Page 1, Line 3, 21, 22, 24, 38, 44; Page 2, Line 49, 58, 75; Page 12, Line 242, 243; Page 13, Line 252, 268, 271, 272, 297).

  • Authors should justify the doses of neferine used. Likewise, in agreement with PK of neferine, also the time of administration should be justified.

About this point, the sentenceThe dose and schedule of administration were chosen based on our pilot study and othersis added in the result section (Page 2, Line 78-79).

  • The method used to solubilize neferine should be better described. How much DMSO was administered for rat? DMSO has demonstrated antioxidant, neuroprotective and cryopreservative effects. Furthermore, it has been demonstrated that DMSO, in a dose dependent manner, has different effects on seizures. Finally, I would like to understand whether the vehicle has any effect.

As suggestion by the reviewer, the sentenceNeferine was dissolved in a saline solution containing DMSO 1% and administrated (30 ml, i.p.) 30 min before KA injection.is added in the result section (Page 14, Line 348-50). In addition, the rats-received DMSO (control group) had no any effect in the present study.

  • Authors have written in methods: “The rats were divided into four groups (5 rats per group)……….”. I have difficult to understand the number of rats used.

In order to make the statement of the sentence more clear, the sentence in the method section is modified to Totally 52 rats were used in this study. The seizure behavior was evaluated in all rats. Immunohistochemistry was performed on fixed brain tissue of 32 rats in four groups (n = 5‒8 rats/group). Fresh brain tissue of 20 rats was used for HPLC and Western blotting in four groups (n = 5 rats/group). (Page 14, Lines 351-355).

  • Authors should report the mortality rate in rats after Kainate injection?

As suggestion by the reviewer, the sentenceIn addition, KA caused 32% mortality of injected rats, and this phenomena was decreased to 15‒18% by neferine pretreatment.is added in the result section (Page 2, Line 85-87).

  • The effect size should also be reported. Likewise, the statistics section should be better described. ANOVA was used to analyze what data? Kruskal-Wallis test was used to analyze what? Why these two tests?

As suggestion by the reviewer, the effect size is added in the result section (Page 2, Line 81, 82; Page 4, Line 104, 105; Page 5, Line 113, 114, 121; Page 7, Line 144- 146; Page 8, Line 170, 171, 179; Page 9, Line 180-182; Page 11, Line 205-207, 226; Page 12, Line 227, 228). In addition, the sentences in the method section are modified to Data were checked for normal distribution using Kolmogorov-Smirnov test. Data for Figure 1 were analyzed using Kruskal-Wallis with Dunn post hoc test. Data for Figure 2-7 were analyzed using one-way analysis of variance (ANOVA) with Tukey’s post hoc test. Eta squared was used to assess the effect size [56]. All analyses were conducted using GraphPad Prism 8 (GraphPad Software, San Diego, California, USA).(Page 16, Line 424-429).

  • Authors in the discussion affirmed that: “this study aimed to evaluate the effect of neferine pretreatment on convulsions during epileptogenesis induced by the glutamate analog KA in rats” As previously reported the epileptogenic period in this modellast about 30 days. By virtue of this, this study did not cover the entire period of epileptogenesis but only the early phase of the latent period.

As suggestion by the reviewer, the sentence in the discussion section is modified to Therefore, this study aimed to evaluate the effect of neferine pretreatment on seizures induced by the glutamate analog KA in rats.(Page 12, Line 244-246).

  • Similarly authors concluded that:” Therefore, neferine could confer beneficial effects against epileptogenesis…..” From my point of view, the experimental design of this study do not permit to draw a similar conclusion.

As suggestion by the reviewer, the sentence in the conclusion section is modified to Therefore, our results demonstrated that neferine might be a potential valuable resource for the prevention and therapy of epilepsy. (Page 16, Line 436-439).

  • I would like to understand why similar effects by different doses of neferine. This should be discussed.

As suggestion by the reviewer, the sentenceIn the present study, neferine, at 10 and 50 mg/kg, exerts similar effects in all analyzed parameters. We infer that the preventive effects of neferine in KA-induced seizure rats might also occur in a lower dose range. Although we did not examine the effect of neferine at the relatively low dose in this study, our finding is consistent with previous studies, which have reported that neferine at a dose of 12–50 mg/kg attenuates ischemia-induced brain damage in rats [18,19,23]is added in the discussion section (Page 14, Line 318-323).

Round 2

Reviewer 2 Report

Authors replied to my comment stating "In pre-exposure to neferine in KA-treated rats, several rats do not exhibit epileptic behavior showing that neferine inhibits epileptogenesis. Therefore, these rats are also be included in subsequent experiments in the present study. In fact, animal epilepsy behavior and brain injury situations do not necessarily correspond. Hope the reviewer can accept our reply."

I still believe that the authors findings were biased by this treatment since it is very well established that epileptogenesis is a function of status epilepticus development and the consequent damage in the model used by the authors (reviewed in Curia et al. 2014). Again, I ask authors to remove from the analysis all the animals which  did not experience convulsive seizures after the kainic acid injection.

Author Response

Response to reviewer 2

ijms-1633899R2

We thank the reviewer for the critical comments and constructive suggestions.

Authors replied to my comment stating "In pre-exposure to neferine in KA-treated rats, several rats do not exhibit epileptic behavior showing that neferine inhibits epileptogenesis. Therefore, these rats are also be included in subsequent experiments in the present study. In fact, animal epilepsy behavior and brain injury situations do not necessarily correspond. Hope the reviewer can accept our reply."

I still believe that the authors findings were biased by this treatment since it is very well established that epileptogenesis is a function of status epilepticus development and the consequent damage in the model used by the authors (reviewed in Curia et al. 2014). Again, I ask authors to remove from the analysis all the animals which did not experience convulsive seizures after the kainic acid injection.

As suggestion by the reviewer, the animals which did not experience convulsive seizures after the kainic acid injection are removed. Therefore, results are corrected in the result section (Page 2, Lines 74-238). In addition, Figure 1A, Figure 1B, Figure 2B, Figure 2C, and Figure 5B are modified.

Reviewer 3 Report

The revised manuscript is acceptable